



# Multi-scale assessment of a grassland productivity model

Shawn D. Taylor[1] and Dawn M. Browning[1]

[1]U.S. Department of Agriculture, Agricultural Research Service, Jornada Experimental Range, New Mexico State University, Las Cruces, New Mexico, United States

**Correspondence:** Shawn D. Taylor (shawn.taylor@usda.gov)

**Abstract.** Grasslands provide many important ecosystem services globally and forecasting grassland productivity in the coming decades will provide valuable information to land managers. Productivity models can be well-calibrated at local scales, but generally have some maximum spatial extent in which they perform well. Here we evaluate a grassland productivity model to find the optimal spatial extent for parameterization, and thus for subsequently applying it in future forecasts for North America. We also evaluated the model on new vegetation types to ascertain its potential generality. We find the model most suitable when incorporating only grasslands, as opposed to also including agriculture and shrublands, and only in the Great Plains and Eastern Temperate Forest ecoregions of North America. The model was not well suited to grasslands in North American Deserts or Northwest Forest ecoregions. It also performed poorly in agriculture vegetation, likely due to management activities, and shrubland vegetation, likely because the model lacks representation of deep water pools. This work allows us to perform long-term forecasts in areas where model performance has been verified, with gaps filled in by future modelling efforts.

## 1 Introduction

Grassland systems span nearly 30% of the global land surface (Adams et al., 1990) and play a prominent role in terrestrial carbon cycles (Parton et al., 2012). Grasslands in North America provide a large proportion of food and fiber agricultural products for the region. Annual productivity of grasslands in central and western North America is driven in large part by precipitation (Sala et al., 2012). Future changes in the amount, intensity, and timing of precipitation will be heterogeneous across North America (Easterling et al., 2017), resulting in heterogeneous changes to grassland productivity. For example, even with consistent shifts in climate, different locations can experience different changes in productivity due to local-scale responses (Zhang et al., 2011; Sala et al., 2012; Knapp et al., 2017). This highlights the need for models which can be resolved at small spatial and temporal scales, thus making long-term grassland productivity forecasts as informative as possible.

There are several potential limitations in the underlying productivity models which can drive such a forecast. Process-based models parameterized with observed data have limited transferability beyond the spatial extent from which their training data came (Taylor et al., 2019). For any location the most accurate model will be one which was parameterized from locally collected data, yet these site-specific models will not generalize to new locations (Basler, 2016). Incorporating more, and diverse,



locations into the model building process will allow it to be more generalizable, yet this comes at a cost of decreased proficiency at all locations (García-Mozo et al., 2008; Basler, 2016). Thus there is an optimal extent in the building and subsequent application of productivity models, which depends on a tradeoff between proficiency at the local scale and applicability at the larger scale.

Here we evaluate a productivity model with the intention of it driving long-term forecasts. The PhenoGrass model developed by Hufkens et al. (2016) is a pulse-response productivity model with temperature and precipitation as the primary drivers. The model is parameterized using observations from the PhenoCam network, which have a small spatial resolution (footprints of < 1ha), sub-daily sampling and sites across all major biomes. These attributes make the PhenoGrass model potentially widely applicable. We expand on the evaluation of the original study by using 84 PhenoCam sites, totalling 89 distinct time series,
with 463 site-years of data. We test the model's performance across varying combinations of North American ecoregions and vegetation types to find an optimal spatial extent in which to parameterize and apply the model. Finally we address where the model performs poorly and how productivity forecasts for these areas could be implemented or improved.

## 2    Methods

### 2.1    PhenoGrass Model

The PhenoGrass model is an ecohydrology model which has interacting state variables for soil water, plant available water, and plant fractional cover (Hufkens et al., 2016). Model inputs are daily precipitation, temperature, potential evapotranspiration (derived from the Hargreaves equation, Hargreaves and Samani (1985)), and solar radiation. The primary output is fractional vegetation cover (fCover). The original model form, derived in Choler et 2010 and Choler et al. 2011, used only temperature and potential evapotranspiration and was parameterized using satellite-derived NDVI data. Hufkens et al. (2016) expanded on
the original Choler model by incorporating growth and senescence restraints from temperature and solar radiation, and also included a scaling factor to convert PhenoCam $G_{cc}$ data to a fractional cover estimate. Hufkens et al. (2016) evaluated the PhenoGrass model using 14 grassland PhenoCam sites across Western North America with a total of 34 site years. They found the modelled fractional cover correlated well with annual productivity at both a daily and annual timescale.

### 2.2    Phenocam Data

The PhenoCam network is a global network of fixed, near-surface cameras capturing true-color images of vegetation throughout the day (Richardson et al., 2018a). Using a ratio of the three RGB bands a greenness metric (green chromatic coordinate, $G_{cc}$) is calculated from each image, resulting in a daily scale time series of canopy greeness. $G_{cc}$ is a unitless metric which is highly correlated with satellite derived NDVI (Richardson et al., 2018b) and flux tower derived primary productivity (Yan et al., 2019; Toomey et al., 2015). Each Phenocam image is subset to one to several different plant vegetation types based on the field of
view. These regions of interest (ROI) serve as the basis for the $G_{cc}$ calculation and subsequent post-processing (Seyednasrollah et al., 2019).





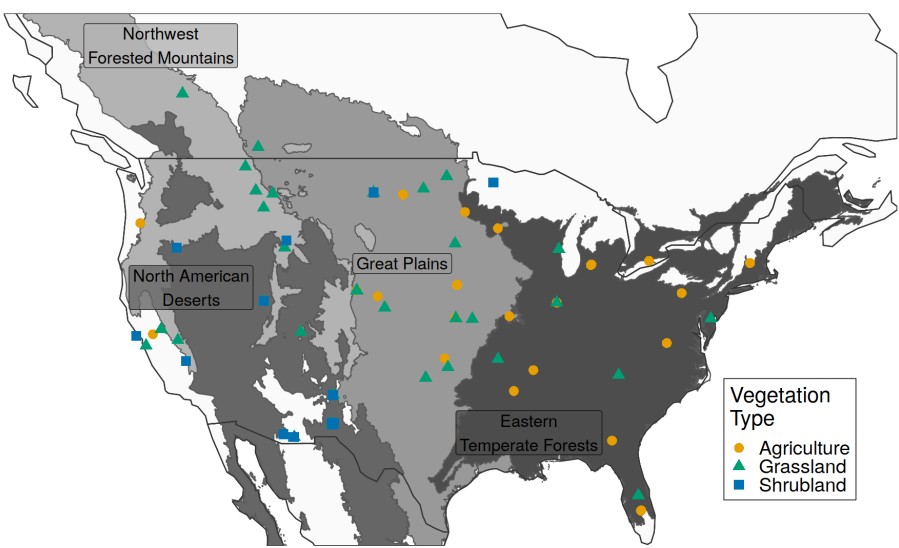

**Figure 1.** Locations of Phenocam sites. Color indicates the vegetation type represented at each site. Vegetation type is defined by the PhenoCam Network. Shading indicates E.P.A. North American Level 1 Ecoregions.

We downloaded all Phenocam data with ROIs of the grasslands (GR), shrublands (SH), and agricultural (AG) vegetation types for the years 2012 to 2018, totalling 89 distinct time series and 463 site-years (Fig. 1, Table A1). As input to the PhenoGrass model we used the 3-day smoothed $G_{cc}$ scaled, for each ROI, from 0-1. In the model parameterization each ROI time series is further transformed to a fractional cover estimate using the local mean annual precipitation (MAP) combined

with a scaling factor (Hufkens et al., 2016; Donohue et al., 2013).

### 2.3   Environmental Data

For historic precipitation and temperature we used the daily 4-km resolution Daymet dataset (Thornton et al., 2018). Climate time series were extracted for the pixel at the location of each phenocam tower. Daily mean temperature was calculated as the average between the Daymet daily minimum and maximum temperature, and smoothed with a 15 day moving average.

Potential evapotranspiration was calculated using the Hargreaves equation (Hargreaves and Samani, 1985). Soil wilting point and field capacity were extracted at each Phenocam location from a global dataset (Global Soil Data Task Group, 2000).

### 2.4   Model Evaluation

To find the most appropriate scale we evaluated the model using three different scales of vegetation type, with 11 total model parameterizations (Fig. 2). The largest scale used all Phenocam locations described above (89 sites). Next were all sites,

respectively, within the three vegetation types indicated by the ROI (grasslands, shrublands, and agricultural). Finally, we



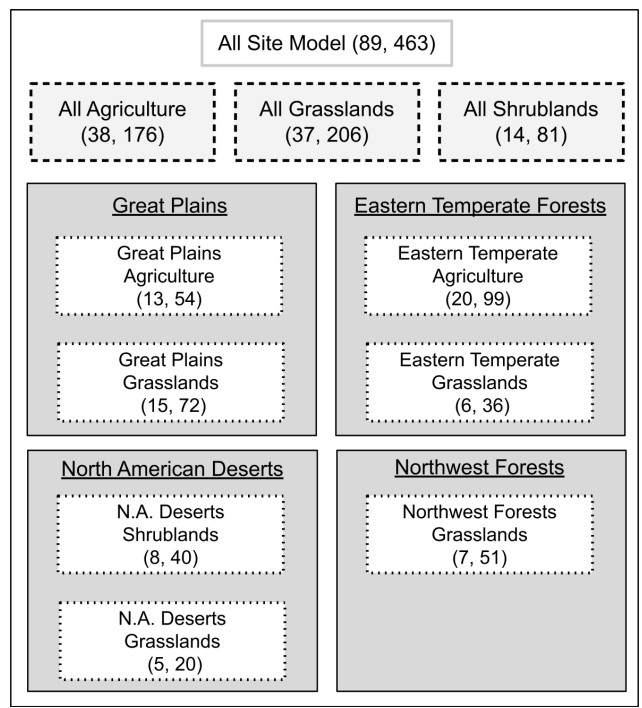

**Figure 2.** Scaling representation of the 11 model parameterizations. Numbers in parentheses represent the number of sites and site years, respectively. Each model uses a different subset of sites ranging from the entire dataset (All Site Model) to one vegetation type within an ecoregion (e.g., Eastern Temperate Forest Grasslands).

parameterized models for each vegetation type within each Level 1 North American Ecoregion (eg. All grassland sites within the Great Plains ecoregion). All sets of parameterized models were limited to have at least five sites.

We evaluated each of the 11 models using the coefficient of determination ($R^2$) and root mean square error (RMSE) of the daily fractional cover estimates. $R^2$ and RMSE were calculated for each site and then averaged across all sites within the respective scale. There was no cross-validation using out of sample data in the initial fitting as it would have been computationally expensive. Rather, error metrics from these in-sample tests were treated as a best case scenario in what each model parameterization can achieve. From these results we used a threshold to select which models to evaluate further using cross-validation. The threshold value was an $R^2$ threshold of 0.65, which is viewed as "acceptable" for time-series models (Ritter and Muñoz-Carpena, 2013).

Models which exceed the threshold were subject to further evaluation. For each model we performed a leave one out cross-validation, where the model was re-fit with one Phenocam site not included in the training data, and then evaluated against this left out site. In this step a scaling coefficient to link mean annual precipitation with PhenoCam $G_{cc}$ was held constant at the value obtained in first fitting. The resulting $R^2$ and RMSE are the average among all modelled sites using their respective out of sample test.




All phenocam data were downloaded using the phenocamr R package (Hufkens et al., 2018). Other packages used in the R 3.6 language were dplyr (Wickham et al., 2017), tidyr (Wickham and Henry, 2018), ggplot2 (Wickham, 2016), daymetr (Hufkens et al., 2018), rgdal (Bivand et al., 2019), and sf (Pebesma, 2018). Python 3.7 packages included scipy (Virtanen et al., 2020), numpy (van der Walt et al., 2011), pandas (McKinney, 2010), and dask (Team, 2016). All code and data used in the analysis is available in the repository at https://github.com/sdtaylor/PhenograssReplication, the PhenoGrass model is implemented in a python package https://github.com/sdtaylor/GrasslandModels. Both are archived permanently on Zenodo (https://doi.org/10.5281/zenodo.3897319).

## 3 Results

At the largest scale, where the PhenoGrass model was parameterized using all 89 sites, the model performed poorly with an average $R^2$ value among sites of 0.31 (Table 1; Fig. 3). Models built using all sites of a respective vegetation type performed poorly as well, though were slightly better than the all site model (Fig. 3). The best model performance was achieved when models were built using a single vegetation type subset to a single ecoregion. Grasslands within the Great Plains and Eastern Temperate Forests ecoregions were the only instances where $R^2$ exceeded the 0.65 threshold, though Grasslands within N.W. Forests came close ($R^2$ = 0.64).

In all 11 iterations the PhenoGrass model tended to underestimate the highest fCover values, and to a lesser degree overpredict the lowest values (Figs. 3,4). The best performing iterations (Grasslands in the Great Plains and Eastern Temperate forests) minimized this effect (Fig. 4). The worst performing iteration, Grasslands in N.A. Deserts, had little variation in predicted fCover values, resulting in the lowest $R^2$ overall.

The grassland vegetation type, subset to specific ecoregions, predominantly outperformed other iterations of the PhenoGrass model (Table 1). Models built using grasslands within the Eastern Temperate Forest and Great Plains ecoregions had the highest average $R^2$ values of 0.82 and 0.69, respectively. Using leave one out cross-validation on these two grassland model iterations resulted in similar errors of 0.79 and 0.67 for the Eastern Temperate Forest and Great Plains, respectively. Though N.W. Forests grasslands had an in-sample $R^2$ just below the 0.65 threshold, the cross-validation was well below it (0.52). Grasslands in the North American deserts were not modelled well at any scale and had the lowest $R^2$ values in the entire analysis. The observed greenness patterns of these desert grasslands had extremely high variability in their magnitude and timing, with short distinct peaks in greenness and numerous off-peak fluctuations. The fitted model, which minimized the mean CVME among the 5 sites, was not able to reproduce this high variability and instead produced fCover values that were severely constrained to a narrow range (Fig. S1).

Agriculture and shrubland sites were poorly modelled at all scales. Performance of agriculture within the E. Temperate Forest ecoregion ($R^2$ = 0.33) improved over the All Agriculture model ($R^2$ = 0.18), but decreased in the Great Plains (from 0.24 to 0.18). There was only a single ecoregion with a minimum of five shrubland sites, N.A. Deserts, and it performed only slightly better than the All Shrubland model. Shrublands in N.A. Deserts did not have the high variability seen in desert grasslands.

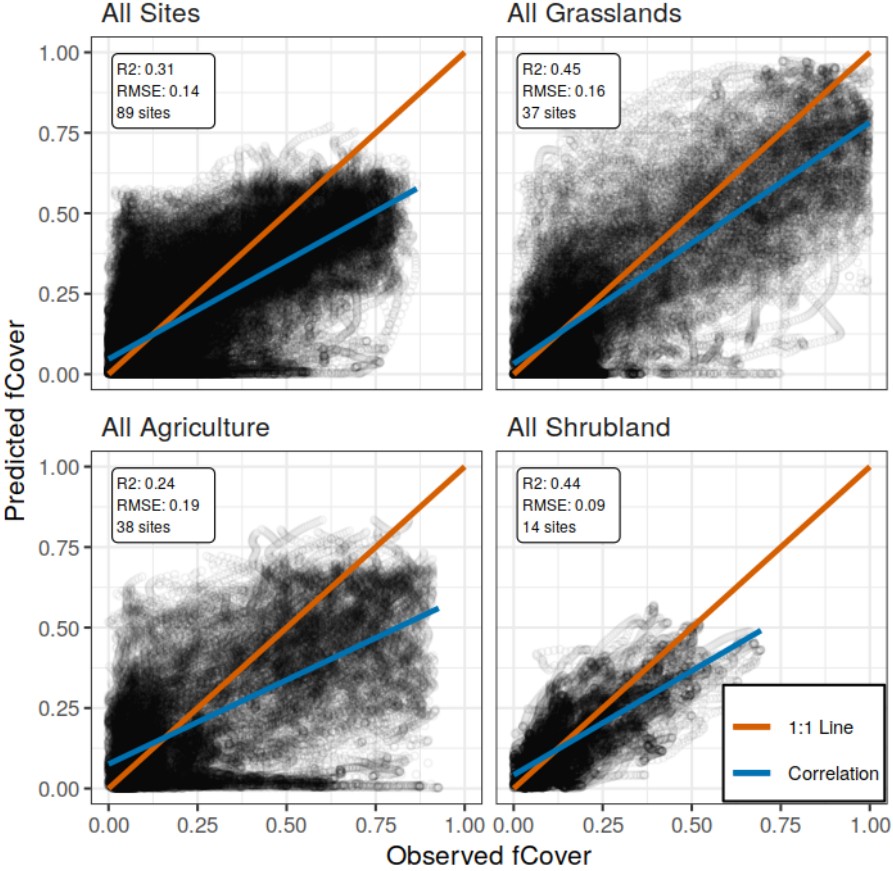

**Figure 3.** Observed and predicted fCover values of the All Site model using all available data, and the three vegetation type models each using all PhenoCam sites with an ROI in the respective vegetation type.

## 4 Discussion

We performed an extensive evaluation of the PhenoGrass model across ecoregions and vegetation types to determine the best scale at which to parameterize and apply the model. We found the model most suitable to grassland vegetation when constrained to the ecoregion level, though it did not perform well in grasslands in the North American desert ecoregion. Shrublands and
5   agriculture were not well represented by the model regardless of the scale. Results from this study will facilitate long-term forecasts of grassland productivity constrained to an appropriate vegetation type and extent.

   The PhenoGrass model performed best in grassland sites embedded within ecoregions. Studies using earlier forms of the model applied it exclusively to grasslands (Choler et al., 2010, 2011; Hufkens et al., 2016), and results here confirm that it performs well in grassland vegetation with two exceptions. The model did not work in the desert grasslands, nor did it generalize
10  well when built using all North American grasslands simultaneously. Grasslands in the N.A. Desert biome coexist with shrubs,

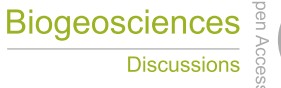

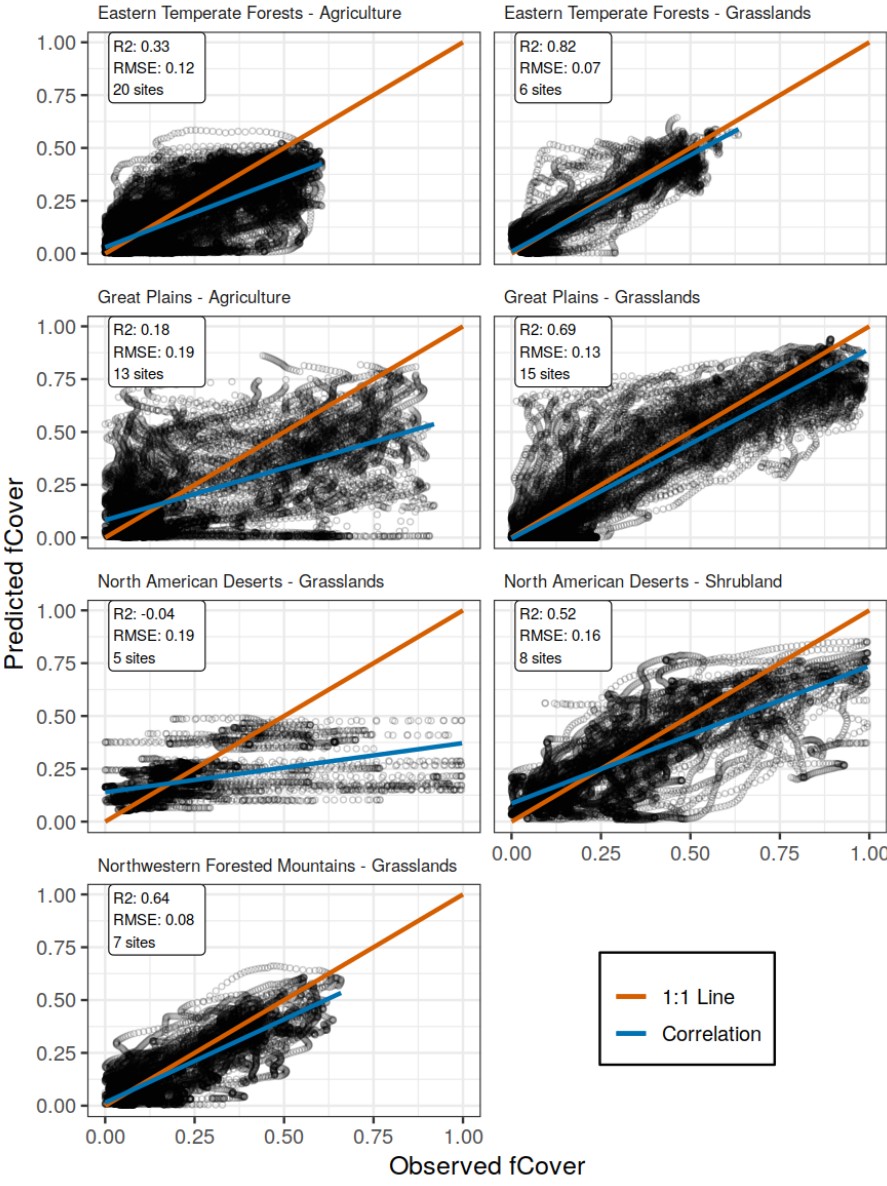

**Figure 4.** Observed and predicted fCover values for seven model iterations where only sites with a specified vegetation type within a single ecoregion were used in model fitting.





**Table 1.** Average site-level coefficient of determination ($R^2$) and root mean square error (RMSE) for each model parameterization. Bold indicates when the $R^2$ was greater than the acceptable threshold of 0.65. Values in parentheses represent the average $R^2$ in leave-1-out cross validation.

|  | $R^2$ | RMSE | Num. Sites | Site Years |
|---|---|---|---|---|
| All Sites | 0.31 | 0.14 | 89 | 462.5 |
| All Agriculture | 0.24 | 0.19 | 38 | 175.7 |
| All Grasslands | 0.45 | 0.16 | 37 | 205.8 |
| All Shrublands | 0.44 | 0.09 | 14 | 81.0 |
| **E. Temperate Forests** |  |  |  |  |
| Agriculture | 0.33 | 0.12 | 20 | 99.2 |
| Grasslands | **0.82 (0.79)** | 0.07 | 6 | 35.7 |
| **Great Plains** |  |  |  |  |
| Agriculture | 0.18 | 0.19 | 13 | 54.3 |
| Grasslands | **0.69 (0.67)** | 0.13 | 15 | 71.5 |
| **N. American Deserts** |  |  |  |  |
| Grasslands | -0.04 | 0.19 | 5 | 20.3 |
| Shrublands | 0.52 | 0.16 | 8 | 39.7 |
| **N.W. Forests** |  |  |  |  |
| Grasslands | 0.64 (0.52) | 0.08 | 7 | 50.7 |

resulting in complex water use dynamics described in more detail below. The pulse-response design of PhenoGrass, which makes it well suited in areas with high cover of perennial grass, is likely not applicable when grasses are interspersed with woody plants.

Shrublands were not well modelled at any scale. Dryland shrubs, representing 8 of the 15 shrubland PhenoCams analysed

5 here, coexist with grasses by accessing different pools of soil water (Weltzin and McPherson, 2000; Muldavin et al., 2008), thus have different responses to precipitation and resulting greenness patterns (Browning et al., 2017; Yan et al., 2019). A prior form of the PhenoGrass model was designed to work with dryland shrubs by using two soil water pools (Ogle and Reynolds, 2004), yet here PhenoGrass, with a single soil water pool, was less effective for shrubland vegetation. The single pool of the PhenoGrass model is coupled with fluxes from precipitation and evapotranspiration, thus is not well-suited for representing the

10 deeper water pools that shrubs can routinely access (Schenk and Jackson, 2002; Ward et al., 2013). Potential improvements would likely need to incorporate a deep soil water pool, in addition to the shallow, which are each utilized by the respective plant functional groups. This has already been implemented in highly parameterized ecohydrology models (Scanlon et al., 2005; Lauenroth et al., 2014) and could potentially be used here to make a more generalized PhenoShrub model to apply across large scales. This approach could also help in modelling N.A. Desert grasslands which coexist among shrubs.



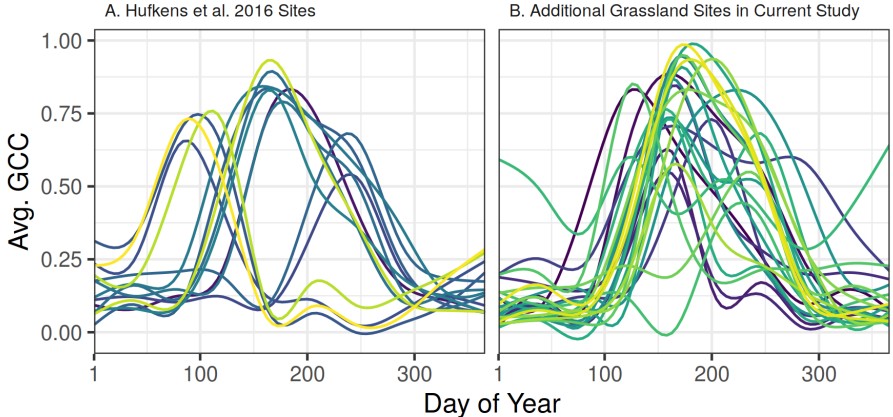

**Figure 5.** Smoothed time series for all 14 grassland sites used in Hufkens et al 2016 (A) and 24 additional grassland sites added in the current study (B). Each line represents the long-term average Green Chromatic Coordinate of a single site across all available years, smoothed using a GAM model.

Agriculture areas performed poorly with the PhenoGrass model. Management practices of crops artificially increase productivity beyond what would naturally occur, and planting and harvest result in abrupt changes in greenness metrics (Bégué et al., 2018). While the results were not necessarily surprising, to our knowledge this is the first attempt to use near-surface images to drive an productivity model for agricultural vegetation. We have shown that the PhenoGrass model, designed for natural systems, does not generalize to actively managed agricultural systems. Future work in using PhenoCam data to model agricultural productivity would likely need to incorporate crop specific parameters and management activity, which other cropland modelling systems use (Fritz et al., 2019). The integration of the PhenoCam network within the Long-Term Agricultural Research (LTAR) will likely be beneficial for this, as the timing and intensity of management activities or experimental treatments can be incorporated into modelling efforts.

Hufkens et al. (2016) originally evaluated the PhenoGrass model using 14 grassland sites distributed among seven North American ecoregions. In their evaluation they had an average $R^2$ of 0.71, while here the model performed poorly when using more than 1 ecoregion. It's likely that the original 14 grassland sites were ideal locations for the PhenoGrass model, since on average they have a single greenup season every year in the spring or summer (Fig. 5A). The additional 24 grassland sites used in the current study have high seasonal variability and elongated growing seasons (Fig. 5B, Fig. S1), and were thus more difficult to represent in a single continental scale grassland model.

## 5 Conclusions

Replication is an important step in the scientific process, especially given newly available data. Here we have validated prior modelling work and highlighted its limitations. Newer small scale vegetation models can be validated in the same framework





and applied to areas where PhenoGrass performs poorly. This can result in a spatial ensemble where the output for any one location and vegetation type is represented by the most appropriate model. Our current work will allow for long-term small scale forecasts of grassland productivity for a large fraction of North America.

*Code and data availability.*   All code and data used in the analysis is available in the repository at https://github.com/sdtaylor/PhenograssReplication,
the PhenoGrass model is implemented in a python package https://github.com/sdtaylor/GrasslandModels. Both are archived permanently on Zenodo (https://doi.org/10.5281/zenodo.3897319).

*Author contributions.*   Shawn concieved of the project. Shawn and Dawn wrote the paper.

*Competing interests.*   The authors declare no competing interests.

*Acknowledgements.*   This research was a contribution from the Long-Term Agroecosystem Research (LTAR) network. LTAR is supported by
the United States Department of Agriculture. The authors acknowledge the USDA Agricultural Research Service (ARS) Big Data Initiative and SCINet high performance computing resources (https://scinet.usda.gov) made available for conducting the research reported in this paper.

   We thank our many collaborators, including site PIs and technicians, for their efforts in support of PhenoCam. The development of PhenoCam has been funded by the Northeastern States Research Cooperative, NSF's Macrosystems Biology program (awards EF-1065029
and EF-1702697), and DOE's Regional and Global Climate Modeling program (award DE-SC0016011).



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
