# Peer review of "Multi-scale assessment of a grassland productivity model"

_Biogeosciences, 2020_

## Referee Comment (RC1) · Anonymous Referee #1 · 13 Oct 2020

The manuscript "Multi-scale assessment of a grassland productivity model" uses the PhenoGrass model to evaluate vegetation cover predictions among sites grouped by ecoregions and vegetation types in North America (focusing on the lower 48 US). The main insights include that the model performs poorly when applied for across ecoregions and across vegetation types, but performs well for non-desert ecoregion x grassland combinations. Phenological studies make important contribution to climate change impact science and also to climate change science by providing long-term observed records. Understanding and modelling what drives phenological patterns in different regions and ecosystems and how these are responding to climate change are important scientific question with direct applications for land management, farming, and forestry. I found the discussion on why the model may have performed poorly for

shrubs or in agricultural settings (8.4-9.15; all numbers refer to page.line) and how the model may be improved particularly interesting and a great addition to the manuscript.

**General comments**

- Data and code are made available online, but I have not tested whether I can re-create the analysis.

- Terminology of forecasts

* page 1.line 1 ("forecasting . . . in the coming decades"), 1.10, 1.20, and throughout: I would welcome a more careful representation of what exactly is implied with forecast claims, e.g., "this work allows us to perform long-term forecasts" (1.10). While I agree that the term "forecast" is used quite generally (e.g., White et al. 2019), it can also be interpreted more specifically for quantitative predictions (e.g., Clark et al. 2001), e.g., weather forecasts. I interpret that the claims made here include time-frames over which a considerable amount of climate change is continuing to occur and thus entail "projections" under specific climate scenarios which then "provide an indication of possibility" instead of "definitive probabilities" (Clark et al. 2001). To make this distinction clear, the climate change community including IPCC do not talk about "predictions"– instead, they are "projections", e.g., quote from the glossary of the AR5 (IPCC 2014 p. 1451): "A climate projection is the simulated response of the climate system to a scenario of future emission or concentration of greenhouse gases and aerosols, generally derived using climate models. Climate projections are distinguished from climate predictions by their dependence on the emission/concentration/radiative forcing scenario used, which is in turn based on assumptions concerning, for example, future socioeconomic and technological developments that may or may not be realized. See also Climate scenario."

* This study does not appear to evaluate/discuss the capability of the model to transfer in time or under climate change type conditions. Thus, the conclusion that "this work allows us to perform long-term forecasts" (1.10) appears to be not based on results

and insights generated by this study.

- Confusing terminology: spatial scale, iteration, spatial extent and spatial resolution (grain) appear to be used as interchangeable, but see, e.g., Wiens 1989. I interpret that the study didn't explicitly assess spatial scale or extent or resolution, but rather differences among ecoregions and ecoregion x vegetation type combinations (e.g., Fig. 2, Table 1). This has implications for the framing, discussion, and conclusions (inclusive title).

- Gaps in method section: no definition or details provided on "parametrization" (e.g., p3.3, p3.14, and throughout)

* Does "parametrization" refer to the "estimation of model parameters" or is it rather how other branches of science use it, e.g., climate scientists as defined in the Glossary to AR5 (IPCC 2014): "technique of representing processes that cannot be explicitly resolved at the spatial or temporal resolution of the model (sub-grid scale processes) by relationships between model-resolved larger-scale variables and the area- or time-averaged effect of such subgrid scale processes."

* I would like to know what was parametrized, e.g., number and types of parameters or processes respectively.

* And it would be valuable to know how this was achieved (estimation method or representing structure respectively). For instance, the result section writes in 5.26 "The fitted model, which minimized the mean CVME among the 5 sites"; I interpret that ecoregion x vegetation-type wide optimization pooled data from 5 sites whereby a metric "CVME" (that is not mentioned or defined elsewhere) was minimized.

- Model evaluation (section 2.4)

* What is the impact of unequal sample size (both number of sites and number of years) among ecoregions (Fig. 2) on model evaluation, particularly based on means across sites? It seems possible that a model may appear to perform worse/better

just by chance in ecoregions with fewer data points. Maybe assess sensitivity with rarefaction?

* How exactly was R2 calculated? It seems that there are a number of different possibilities (4.3).

* I don't believe that RMSE (4.3) is a very useful metric to compare different datasets (e.g., comparisons among ecoregions) because mean cover values differ among regions (see Fig. 3). For instance, a RMSE of 0.09 for shrublands cannot be directly compared to the RMSE of 0.16 for grasslands if grasslands have on average higher cover than shrublands. Unfortunately, mean cover values are not reported (e.g., they could be added to Table 1). A normalized form of RMSE (e.g., the coefficient of variation of RMSE = RMSE / mean) would make among-region comparisons easier to interpret.

* Why was the estimate of the "scaling coefficient" not part of the cross-evaluation (4.12)? Depending on the sensitivity of the model to this parameter, the "out-of-sample" estimate may be considerably influenced (as in augmented) by this estimate based on all data.

* I agree that mean R2 and mean RMSE among sites may provide a reasonable estimate of average performance across sites (4.14); however, use cases at specific sites would likely need to also consider expected worst case performance. For instance, Fig. 3 hints at (not possible to know for sure because point density is not shown) that a large number of data points are predicted at 0 cover irrespective of the observed cover value (all but "All Shrubland" model) – this could be driven by to one or a few poorly performing site or by some years, etc.

* A blue line is shown and labelled "Correlation" in Fig. 3. This looks rather like a simple linear regression line than a "correlation" (point estimate)? Was this regression the basis to estimate R2 (see comment to 4.3)?

* Result section (5.8ff): The entire result section focuses on R2 only and does not report on RMSE results – despite method section and figures.

**Specific comments**

- It is not clear to me why the manuscript argues that the theoretical expectation is that process-based transfer in general worse to new conditions than others, e.g., statistical models (1.21f.). It seems that at least several authors have argued for exactly the opposite expectation, e.g., Grimm and Berger 2016, Radchuk et al. 2019, while others have pointed out real-world limitations while maintaining the theoretical expectation, e.g., that process-based models are often limited because they require large amounts of data and a complete understanding of relevant processes, e.g., Pennekamp et al. 2017, Yates et al. 2018, Bouchet et al. 2019).

- 1.23-2.4 appear to refer to the problematic of overfitting versus building a generalizable model without referring to relevant literature and without defining what exactly is meant by "most accurate". It seems that approaches to minimize overfitting have been frequently discussed in the literature including parsimony (e.g., likelihood-based or information theoretic approaches).

- 2.18: Maybe clarify whether "fractional vegetation cover" includes all vegetation cover combined (whether grasses or not) or estimates are produced separate for grass, shrub, and "agricultural" vegetation types (as hinted at in 3.1)

- 3.7: It seems that the original Daymet product has a 1-km resolution. Is this a typo or was it aggregated here to 4-km. If the latter, explain why and how.

- 3.7: "Climate time series": daily meteorological time series data rather sound like weather data to me

- 3.11: Why use a 20+ year old global soil dataset instead of using one of the many updated and improved ones, e.g., WISE-based HWSD (Batjes 2016), SoilGrids (Hengl et al. 2017) etc., and why not use a regional dataset, e.g., gNATSGO (NRCS 2020),

POLARIS (Chaney et al. 2019), etc.? How sensitive is the PhenoGrass model to differences in soil variables that occur typically between Global Soil Data Task Group (2000) and others?

- 5.15: Why are the different model fits/parametrizations now suddenly called "iterations"? This is confusing because "iteration" can have a specific and different meaning in parameter estimation/model fitting than what appears to be implied here.

- Fig. 3: Explain what the individual data points are, days pooled from all sites and years?

**References - Batjes, N. H. 2016. Harmonized soil property values for broadscale modelling (WISE30sec) with estimates of global soil carbon stocks. Geoderma 269:61–68. https://doi.org/10.1016/j.geoderma.2016.01.034.**

- Bouchet, P. J., A. T. Peterson, D. Zurell, C. F. Dormann, D. Schoeman, R. E. Ross, P. Snelgrove, A. M. M. Sequeira, M. J. Whittingham, L. Wang, G. Rapacciuolo, S. Oppel, C. Mellin, V. Lauria, P. K. Krishnakumar, A. R. Jones, S. Heinänen, R. K. Heikkinen, E. J. Gregr, A. H. Fielding, M. J. Caley, A. M. Barbosa, A. J. Bamford, H. Lozano-Montes, S. Parnell, S. Wenger, and K. L. Yates. 2019. Better Model Transfers Require Knowledge of Mechanisms. Trends in Ecology & Evolution 34:489–490. https://doi.org/10.1016/j.tree.2019.04.006.

- Chaney, N. W., B. Minasny, J. D. Herman, T. W. Nauman, C. Brungard, C. L. S. Morgan, A. B. McBratney, E. F. Wood, and Y. T. Yimam. 2019. POLARIS soil properties: 30-meter probabilistic maps of soil properties over the contiguous United States. Water Resources Research 55:2916–2938. https://doi.org/10.1029/2018WR022797.

- Clark, J. S., S. R. Carpenter, M. Barber, S. Collins, A. Dobson, J. A. Foley, D. M. Lodge, M. Pascual, R. Pielke, W. Pizer, C. Pringle, W. V. Reid, K. A. Rose, O. Sala, W. H. Schlesinger, D. H. Wall, and D. Wear. 2001. Ecological Forecasts: An Emerging Imperative. Science 293:657–660. https://doi.org/10.1126/science.293.5530.657.

- Grimm, V., and U. Berger. 2016. Structural realism, emergence, and predictions in next-generation ecological modelling: Synthesis from a special issue. Ecological Modelling 326:177–187. https://doi.org/10.1016/j.ecolmodel.2016.01.001.

- Hengl, T., J. M. de Jesus, G. B. M. Heuvelink, M. R. Gonzalez, M. Kilibarda, A. Blagotić, W. Shangguan, M. N. Wright, X. Geng, B. Bauer-Marschallinger, M. A. Guevara, R. Vargas, R. A. MacMillan, N. H. Batjes, J. G. B. Leenaars, E. Ribeiro, I. Wheeler, S. Mantel, and B. Kempen. 2017. SoilGrids250m: Global gridded soil information based on machine learning. PLOS ONE 12:e0169748. https://doi.org/10.1371/journal.pone.0169748.

- IPCC. 2014. Climate change 2013: the physical science basis. Contribution of working group I to the fifth assessment report of the Intergovernmental Panel on Climate Change. 1535 pages. Cambridge University Press, Cambridge, United Kingdom and New York, NY, USA.

- Pennekamp, F., M. W. Adamson, O. L. Petchey, J.-C. Poggiale, M. Aguiar, B. W. Kooi, D. B. Botkin, and D. L. DeAngelis. 2017. The practice of prediction: What can ecologists learn from applied, ecology-related fields? Ecological Complexity 32:156–167. https://doi.org/10.1016/j.ecocom.2016.12.005.

- Radchuk, V., S. Kramer-Schadt, and V. Grimm. 2019. Transferability of Mechanistic Ecological Models Is About Emergence. Trends in Ecology & Evolution 34:487–488. https://doi.org/10.1016/j.tree.2019.01.010.

- Soil Survey Staff. 2020. Gridded National Soil Survey Geographic (gNATSGO) Database for the Conterminous United States. Available online at https://nrcs.app.box.com/v/soils. (FY2020 official release). United States Department of Agriculture, Natural Resources Conservation Service.

- White, E. P., G. M. Yenni, S. D. Taylor, E. M. Christensen, E. K. Bledsoe, J. L. Simonis, and S. K. M. Ernest. 2019. Developing an automated iterative near-term forecasting system for an ecological study. Methods in Ecology and Evolution 10:332–344. https://doi.org/10.1111/2041-210X.13104.

- Wiens, J. A. 1989. Spatial Scaling in Ecology. Functional Ecology 3:385–397.

- Yates, K. L., P. J. Bouchet, M. J. Caley, K. Mengersen, C. F. Randin, S. Parnell, A. H. Fielding, A. J. Bamford, S. Ban, A. M. Barbosa, C. F. Dormann, J. Elith, C. B. Embling, G. N. Ervin, R. Fisher, S. Gould, R. F. Graf, E. J. Gregr, P. N. Halpin, R. K. Heikkinen, S. Heinänen, A. R. Jones, P. K. Krishnakumar, V. Lauria, H. Lozano-Montes, L. Mannocci, C. Mellin, M. B. Mesgaran, E. Moreno-Amat, S. Mormede, E. Novaczek, S. Oppel, G. Ortuño Crespo, A. T. Peterson, G. Rapacciuolo, J. J. Roberts, R. E. Ross, K. L. Scales, D. Schoeman, P. Snelgrove, G. Sundblad, W. Thuiller, L. G. Torres, H. Verbruggen, L. Wang, S. Wenger, M. J. Whittingham, Y. Zharikov, D. Zurell, and A. M. M. Sequeira. 2018. Outstanding Challenges in the Transferability of Ecological Models. Trends in Ecology & Evolution 33:790–802. https://doi.org/10.1016/j.tree.2018.08.001.

---

## Referee Comment (RC2) · Anonymous Referee #2 · 27 Oct 2020

Using the observed data at the sites of phenoCam network, the authors evaluated the performance of a productivity model, PhenoGrass at different ecosystem types. They identified the 'optimal spatial extent', in which the model performed the best. I have several major concerns on the manuscript, which I think are very important before the publication of this paper. 1. Apparently, this study just evaluated the performance of a model, identifying which ecosystem types the model perform best. However, this evaluation did not fill any knowledge gap on the way of improving our capability of forecasting. 2. The model results suggest that the model perform best in grassland ecosystems. I can guess that is within expectation, because it is likely that the model was originally developed for grassland ecosystems according to its name, PhenoGrass. No explanation was provided on how the model has been updated on simulating pro-

ductivity in other ecosystem types. 3. Here the evaluation focus on primary productivity. Why not use the GPP data observed at fluxnet sites by eddy covariance towers, but the fcover at phnenoCam sites? 4. More text is needed to elaborate the principle of the model. Key equations are needed as appendix. 5. How the parameters of the model were determined? How the parameters varied across ecosystem types? 6. How to use the image data (RBG) to estimate fcover? Is there some uncertainty at this step?

---

## Author Comment (AC1) · 17 Nov 2020

article [utf8]inputenc

**Referee 2 Comments are in plain text, with Author responses in bold**

Using the observed data at the sites of phenoCam network, the authors evaluated the performance of a productivity model, PhenoGrass at different ecosystem types. They identified the 'optimal spatial extent', in which the model performed the best. I have several major concerns on the manuscript, which I think are very important before the publication of this paper.

1. Apparently, this study just evaluated the performance of a model, identifying which

ecosystem types the model perform best. However, this evaluation did not fill any knowledge gap on the way of improving our capability of forecasting.

**The original model was already used for long term projections of grassland productivity in the highly cited Hufkens et al. 2016 paper. We feel that re-evaluating that model with newer and more extensive data (featuring 89 sites and 463 site-years) to examine performance and identify limitations on where it is applicable is a valid contribution.**

2. The model results suggest that the model perform best in grassland ecosystems. I can guess that is within expectation, because it is likely that the model was originally developed for grassland ecosystems according to its name, PhenoGrass. No explanation was provided on how the model has been updated on simulating productivity in other ecosystem types.

**Despite its name the PhenoGrass model has no component which is specific to grass. The primary response variable, fractional vegetation cover, can theoretically apply to any vegetation type. The original formulation derived in Ogle Reynolds 2004 was used, with hypothetical parameters, on several plant functional types including annual and perennial grasses, cacti, and shrubs. Choler et al. 2010, 2011 modified this formulation and modeled grasslands using NDVI, since grasslands are a homogenous functional type which have a distinct NDVI signal and a well studied relationship with precipitation. Hufkens et al. 2016 expanded on the Choler 2011 model by adding temperature and daylength constraints, but again nothing specific to grasslands. Because PhenoCams allow us to isolate the annual growth of specific vegetation types at a daily scale, it was worth evaluating the PhenoGrass model on other vegetation types.**

**We will revise text in the methods to include the following clarification on this:**

**"Despite its name the PhenoGrass model can theoretically apply to any vegeta-**

tion type with a distinct growth signal in response to precipitation, as hypothesized in the original threshold-delay model (Ogle  Reynolds 2004). Here we use two other vegetation types, shrubs and agricultural plots, to test how applicable it is beyond grasslands."

3. Here the evaluation focus on primary productivity. Why not use the GPP data observed at fluxnet sites by eddy covariance towers, but the fcover at phenoCam sites?

**The PhenoGrass model is designed to work with fractional vegetation cover, the proportion of ground covered by live vegetation. Flux tower measurements cannot quantify this, thus they could not be used.**

4. More text is needed to elaborate the principle of the model. Key equations are needed as appendix.
**The model is fully described in Hufkens et al. 2016 and we made no modifications to it for this study. On revision we will include the full equations in the appendix for clarity.**

5. How the parameters of the model were determined? How the parameters varied across ecosystem types?

**We will include the following main text in the method to clarify this.**

**"Parameterization was done using differential evolution, a global optimization algorithm, to minimize the mean coefficient of variation of the mean absolute error (F), which accounts for variation among average Gcc values among sites (Choler et al. 2011).**

$$F = \frac{1}{N} \sum_{j=1}^{N} CVMAE_j$$

$$CVMAE_j = \frac{\frac{1}{i}\sum_{i=1}^{n}|fCover_{i,obs} - fCover_{i,pred}|}{\overline{fCover_{obs}}}$$

**Where N is the number of sites, i is the number of daily values in each site, $fCover_{i,obs}$ and $fCover_{i,pred}$ are observed and predicted values, respectively. $\overline{fCover_{obs}}$ is the average fCover at each site."**

**We will also include the following table in the appendix describing the final parameter values for the two models which met the threshold.**

| parameter | Great Plains | E. Temperate Forests |
|-----------|-------------|----------------------|
| b2 | 0.0021756 | 0.0143214 |
| b3 | 0.0607134 | 0.0269089 |
| b4 | 0.2630305 | 9.8632117 |
| Phmax | 48.1106340 | 49.6940973 |
| Phmin | 25.5471298 | 33.1345876 |
| Topt | 29.7376494 | 35.6335041 |
| L | 3.0685254 | 3.1769617 |
| h | 10.3601586 | 949.5914722 |
| mean_cvmae | 0.3882550 | 0.2734598 |

6. How to use the image data (RBG) to estimate fcover? Is there some uncertainty at this step?

**Phenocam images are subset to a region of interest (ROI) which isolates a specific vegetation type in the camera field of view. RGB values from within each ROI are converted to the green chromatic coordinate (Gcc):**

**Gcc = G / (R + G + B)**

**Where R,G, and B are the average digital number values of the respective color within the ROI. This produces a daily normalized greenness index which tracks vegetation extremely well (Richardson et al. 2018). Gcc values are converted to fCover via asymptomatic transfer function**

**fCover = Gcc * S**

**S = MAP/ (MAP + h)**

**Where MAP is the mean annual precipitation at a site and h is a parameter estimated along with the rest of phenograss model parameters (Hufkens et al. 2016).**

**We will include these details in the appendix along with the full model description. Uncertainty around the Gcc is minimized by using the 90th percentile of the 3-day moving average, which reduces the effects of different lighting conditions.**

References:

Richardson, A. D., Hufkens, K., Milliman, T., Aubrecht, D. M., Chen, M., Gray, J. M., . . . Frolking, S. (2018). Tracking vegetation phenology across diverse North American biomes using

PhenoCam imagery. Scientific Data, 5(1), 180028. https://doi.org/10.1038/sdata.2018.28

Hufkens, K., Keenan, T. F., Flanagan, L. B., Scott, R. L., Bernacchi, C. J., Joo, E., . . . Richardson, A. D. (2016). Productivity of North American grasslands is increased under future climate scenarios despite rising aridity. Nature Climate Change, 6(7), 710–714. https://doi.org/10.1038/nclimate2942

Choler, P., Sea, W., Leuning, R. (2011). A Benchmark Test for Ecohydrological Models of Interannual Variability of NDVI in Semi-arid Tropical Grasslands. Ecosystems, 14(2), 183–197. https://doi.org/10.1007/s10021-010-9403-9

Choler, P., Sea, W., Briggs, P., Raupach, M., Leuning, R. (2010). A simple ecohydrological model captures essentials of seasonal leaf dynamics in semi-arid tropical grasslands. Biogeosciences, 7(3), 907–920. https://doi.org/10.5194/bg-7-907-2010

Ogle, K., Reynolds, J. F. (2004). Plant responses to precipitation in desert ecosystems: integrating functional types, pulses, thresholds, and delays. Oecologia, 141(2), 282–294. https://doi.org/10.1007/s00442-004-1507-5

---

## Author Comment (AC2) · 17 Nov 2020

article [utf8]inputenc

**Referee 1 Comments are in plain text, with Author responses in bold**

The manuscript "Multi-scale assessment of a grassland productivity model" uses the PhenoGrass model to evaluate vegetation cover predictions among sites grouped by ecoregions and vegetation types in North America (focusing on the lower 48 US). The main insights include that the model performs poorly when applied for across ecoregions and across vegetation types, but performs well for non-desert ecoregion x

grassland combinations. Phenological studies make important contribution to climate change impact science and also to climate change science by providing long-term observed records. Understanding and modelling what drives phenological patterns in different regions and ecosystems and how these are responding to climate change are important scientific question with direct applications for land management, farming, and forestry. I found the discussion on why the model may have performed poorly for shrubs or in agricultural settings (8.4-9.15; all numbers refer to page.line) and how the model may be improved particularly interesting and a great addition to the manuscript.

General comments - Data and code are made available online, but I have not tested whether I can re-create the analysis.

- Terminology of forecasts * page 1.line 1 ("forecasting . . . in the coming decades"), 1.10, 1.20, and throughout: I would welcome a more careful representation of what exactly is implied with forecast claims, e.g., "this work allows us to perform long-term forecasts" (1.10). While I agree that the term "forecast" is used quite generally (e.g., White et al. 2019), it can also be interpreted more specifically for quantitative predictions (e.g., Clark et al. 2001), e.g., weather forecasts. I interpret that the claims made here include time-frames over which a considerable amount of climate change is continuing to occur and thus entail "projections" under specific climate scenarios which then "provide an indication of pos- sibility" instead of "definitive probabilities" (Clark et al. 2001). To make this distinction clear, the climate change community including IPCC do not talk about "predictions"– instead, they are "projections", e.g., quote from the glossary of the AR5 (IPCC 2014 p. 1451): "A climate projection is the simulated response of the climate system to a sce- nario of future emission or concentration of greenhouse gases and aerosols, generally derived using climate models. Climate projections are distinguished from climate pre- dictions by their dependence on the emission/concentration/radiative forcing scenario used, which is in turn based on assumptions concerning, for example, future socioeconomic and technological developments that may or may not be realized. See also Climate scenario."

**We will change the term "forecast" to "projection" in the text.**

\* This study does not appear to evaluate/discuss the capability of the model to transfer in time or under climate change type conditions. Thus, the conclusion that "this work allows us to perform long-term forecasts" (1.10) appears to be not based on results and insights generated by this study.

**It's true that we do not test model transferability across time. With a median length of 4.3 years per site we do not feel the current dataset has ample length for a proper temporal out of sample test. Validation of models under current conditions is the 1st step toward applying them toward climate projections though, thus the above statement is still valid.**

**We'll include the following caveat at the end in the discussion to highlight the importance of model transferability in time.**

**"This highlights the need for longer time series in evaluating small scale models as it may take several years for a single location to experience the full range of variability. As PhenoCam data collection continues then temporally out of sample validation can be done to better model performance into novel conditions."**

- Confusing terminology: spatial scale, iteration, spatial extent and spatial resolution (grain) appear to be used as interchangeable, but see, e.g., Wiens 1989. I interpret that the study didn't explicitly assess spatial scale or extent or resolution, but rather differences among ecoregions and ecoregion x vegetation type combinations (e.g., Fig. 2, Table 1). This has implications for the framing, discussion, and conclusions

(inclusive title).

**We admit this can be confusing. There is a clear distinction between 1) using all available sites versus 2) only sites within a specific ecoregion, as distinct spatial extents, but vegetation types are not described well using these terms. We settled by using the term "spatial scale" throughout since it's viewed as more generic than "spatial resolution/grain" or "spatial extent". On revision we will include the following text to explicitly state the definition, used here, for "spatial scale" as the combination of different ecoregions and vegetation types.**

**"Here we use the term "spatial scale" to refer to the combination of ecoregion/s and vegetation type/s used within each model. This includes using all vegetation types with an ecoregion, or all sites of a specific vegetation type from several ecoregions"**

**We'll also remove all mentions of extent when discussing our own results. Our use of "resolution" was only used to describe the daymet and phenocam data attributes, thus we will keep those in place.**

- Gaps in method section: no definition or details provided on "parametrization" (e.g.,p3.3, p3.14, and throughout)

* Does "parametrization" refer to the "estimation of model parameters" or is it rather how other branches of science use it, e.g., climate scientists as defined in the Glossary to AR5 (IPCC 2014): "technique of representing processes that cannot be explicitly resolved at the spatial or temporal resolution of the model (sub-grid scale processes) by relationships between model-resolved larger-scale variables and the area- or time-averaged effect of such subgrid scale processes."

* I would like to know what was parametrized, e.g., number and types of parameters or processes respectively.

* And it would be valuable to know how this was achieved (estimation method or representing structure respectively). For instance, the result section writes in 5.26 "The fitted model, which minimized the mean CVME among the 5 sites"; I interpret that ecoregion x vegetation-type wide optimization pooled data from 5 sites whereby a metric "CVME" (that is not mentioned or defined elsewhere) was minimized.

**Parameterization is indeed estimating the 8 model parameters via minimizing a loss function. We will include the following main text in the method to clarify this.**

**"Parameterization was done using differential evolution, a global optimization algorithm, to minimize the mean coefficient of variation of the mean absolute error (F), which accounts for variation among average Gcc values among sites (Choler et al. 2011).**

$$F = \frac{1}{N} \sum_{j=1}^{N} CVMAE_j$$

$$CVMAE_j = \frac{\frac{1}{i} \sum_{i=1}^{n} |fCover_{i,obs} - fCover_{i,pred}|}{\overline{fCover_{obs}}}$$

**Where N is the number of sites, i is the number of daily values in each site, $fCover_{i,obs}$ and $fCover_{i,pred}$ are observed and predicted values, respectively. $\overline{fCover_{obs}}$ is the average fCover at each site."**

**The PhenoGrass model is described fully in Hufkens et al. 2016 and we used it here without modification, but we will include the full model description, including equations, in the appendix for clarity.**

[Figure]

We will also include the following table in the appendix describing the final parameter values for the two models which met the threshold.

| parameter | Great Plains | E. Temperate Forests |
|---|---|---|
| b2 | 0.0021756 | 0.0143214 |
| b3 | 0.0607134 | 0.0269089 |
| b4 | 0.2630305 | 9.8632117 |
| Phmax | 48.1106340 | 49.6940973 |
| Phmin | 25.5471298 | 33.1345876 |
| Topt | 29.7376494 | 35.6335041 |
| L | 3.0685254 | 3.1769617 |
| h | 10.3601586 | 949.5914722 |
| mean_cvmae | 0.3882550 | 0.2734598 |

- Model evaluation (section 2.4)

* What is the impact of unequal sample size (both number of sites and number of years) among ecoregions (Fig. 2) on model evaluation, particularly based on means across sites? It seems possible that a model may appear to perform worse/better just by chance in ecoregions with fewer data points. Maybe assess sensitivity with rarefaction?

**Different sample sizes and time series lengths were accounted for in two ways. The loss function, the mean CVMAE, gave each site the same weight regardless of time series length. The evaluation metric, $R^2$, is also robust against sample size differences as long as sample sizes are not extremely low (McCuen et. al 2006).**

**McCuen, Richard H., Zachary Knight, and A. Gillian Cutter. "Evaluation of the Nash–Sutcliffe efficiency index." Journal of hydrologic engineering 11.6 (2006):**

**597-602.**

* How exactly was R2 calculated? It seems that there are a number of different possibilities (4.3).

**This is the coefficient of determination and it has the following equation:**

$$R^2 = 1 - \frac{\sum_{i=1}^{n}(fCover_{obs} - fCover_{pred})^2}{\sum_{i=1}^{n}(fCover_{obs} - \overline{fCover_{obs}})^2}$$

**Where n is the number of observations for a single site, and $\overline{fCover_{obs}}$ is the mean fCover for the site. As opposed to a regression $R^2$, this metric uses observed versus predicted values in relation to the 1:1 line. It's common in the ecological literature, but in the hydrology literature this same equation is known as the Nash Sutcliffe coefficient of efficiency (NSE, see Ritter Muñoz-Carpena, 2013). On revision we will remove $R^2$ from the manuscript and replace it with NSE, with the above equation and definition, to avoid any confusion. We will also emphasize the reported results as the mean NSE across sites (eg. $\overline{NSE}$).**

**Ritter, A., Muñoz-Carpena, R. (2013). Performance evaluation of hydrological models: Statistical significance for reducing subjectivity in goodness-of-fit assessments. Journal of Hydrology, 480, 33–45. https://doi.org/10.1016/j.jhydrol.2012.12.004**

* I don't believe that RMSE (4.3) is a very useful metric to compare different datasets (e.g., comparisons among ecoregions) because mean cover values differ among regions (see Fig. 3). For instance, a RMSE of 0.09 for shrublands cannot be directly compared to the RMSE of 0.16 for grasslands if grasslands have on average higher cover than shrublands. Unfortunately, mean cover values are not reported (e.g., they

could be added to Table 1). A normalized form of RMSE (e.g., the coefficient of variation of RMSE = RMSE / mean) would make among-region comparisons easier to interpret.

**On revision we will replace RMSE in the results with the mean CVMAE, which is the loss function and is normalized to the within site variation (see above).**

\* Why was the estimate of the "scaling coefficient" not part of the cross-evaluation (4.12)? Depending on the sensitivity of the model to this parameter, the "out-of-sample" estimate may be considerably influenced (as in augmented) by this estimate based on all data.

**Holding this value constant in the cross-validation step is how the model was originally evaluated in Hufkins et al. 2016, which we attempted to replicate as much as possible.**

\* I agree that mean R2 and mean RMSE among sites may provide a reasonable estimate of average performance across sites (4.14); however, use cases at specific sites would likely need to also consider expected worst case performance. For instance, Fig. 3 hints at (not possible to know for sure because point density is not shown) that a large number of data points are predicted at 0 cover irrespective of the observed cover value (all but "All Shrubland" model) – this could be driven by to one or a few poorly performing site or by some years, etc.

**This is correct. The poorly performing models in Fig. 3 were noted as such since they did not meet the threshold for further evaluation.**

\* A blue line is shown and labelled "Correlation" in Fig. 3. This looks rather like a simple linear regression line than a "correlation" (point estimate)? Was this regression

the basis to estimate R2 (see comment to 4.3)?

**The blue is indeed the regression line, but labelling it as such would give the false impression that the y-axis values are a function of the x-axis value. Here the figure is showing predicted versus observed results, where a perfect prediction is shown by the 1:1 line. The blue "correlation" line is meant to indicate an overall trend relative to a perfect model fit. See above for the equation for $R^2$.**

\* Result section (5.8ff): The entire result section focuses on R2 only and does not report on RMSE results – despite method section and figures.

**RMSE will be removed from results and replaced with the mean CVMAE (see above).**

Specific comments

- It is not clear to me why the manuscript argues that the theoretical expectation is that process-based transfer in general worse to new conditions than others, e.g., statistical models (1.21f.). It seems that at least several authors have argued for exactly the opposite expectation, e.g., Grimm and Berger 2016, Radchuk et al. 2019, while others have pointed out real-world limitations while maintaining the theoretical expectation, e.g., that process-based models are often limited because they require large amounts of data and a complete understanding of relevant processes, e.g., Pennekamp et al. 2017, Yates et al. 2018, Bouchet et al. 2019).

- 1.23-2.4 appear to refer to the problematic of overfitting versus building a generalizable model without referring to relevant literature and without defining what exactly is meant by "most accurate". It seems that approaches to minimize overfitting have been frequently discussed in the literature including parsimony (e.g., likelihood-based or information theoretic approaches).

**Where the PhenoGrass model falls in the process vs. statistical model definition is ambiguous, since it has elements of both. From a process model perspective it has elements for a soil water pool and evapotranspiration, but also parameters for transpiration which are essentially statistical coefficients. Because of the latter element it is susceptible to overfitting to local conditions, thus transferability to new locations is limited. On revision we will describe PhenoGrass as a "low dimensional model" to better reflect this.**

**Our analysis here cannot be approached with traditional model selection approaches. Given the problem that the phenograss model can fit very well to local conditions, but can only generalize so far beyond that, we set out to find an optimal scale at which it could be parameterized.**

**Following is new text to replace the the current 2nd introduction paragraph to reflect the above comments:**

**". . . This highlights the need for models which can be resolved at small spatial and temporal scales, thus making projections of grassland productivity as informative as possible.**

**A promising method is low dimensional models, which are process models with some simplified components (Choler et al. 2010, 2011). For example, a low dimensional model might approximate transpiration to a function of potential evapotranspiration, soil available water, and live vegetation cover along with a single parameter. As opposed to a high dimensional model with multiple functions accounting for leaf area index, stomatal conductance, rooting depth and surface area, etc. (Caylor et al. 2009, Asbjornsen et al. 2011). The low dimensional model is advantageous since it can generalize across broad regions with relatively few inputs. Yet they are still susceptible to over-fitting to local conditions since parameters or model structure can be tied to specific locations or plant functional groups (Fisher Koven 2020). Thus parameterizing low-dimensional**

**models must be done with care such that they are applicable to a broad area while maintaining an acceptable level of accuracy.**

**Here we evaluate a low-dimensional model with the intention of it driving climate projections. The PhenoGrass model developed by Hufkens et al. (2016) ..... "**

Asbjornsen, H., Goldsmith, G. R., Alvarado-Barrientos, M. S., Rebel, K., Van Osch, F. P., Rietkerk, M., . . . Dawson, T. E. (2011). Ecohydrological advances and applications in plant-water relations research: a review. Journal of Plant Ecology, 4(1–2), 3–22. https://doi.org/10.1093/jpe/rtr005

Caylor, K. K., Scanlon, T. M., Rodriguez-Iturbe, I. (2009). Ecohydrological optimization of pattern and processes in water-limited ecosystems: A trade-off-based hypothesis. Water Resources Research, 45(8), 1–15. https://doi.org/10.1029/2008WR007230

Choler, P., Sea, W., Briggs, P., Raupach, M., Leuning, R. (2010). A simple ecohydrological model captures essentials of seasonal leaf dynamics in semi-arid tropical grasslands. Biogeosciences, 7(3), 907–920. https://doi.org/10.5194/bg-7-907-2010

Choler, P., Sea, W., Leuning, R. (2011). A Benchmark Test for Ecohydrological Models of Interannual Variability of NDVI in Semi-arid Tropical Grasslands. Ecosystems, 14(2), 183–197. https://doi.org/10.1007/s10021-010-9403-9

Fisher, R. A., Koven, C. D. (2020). Perspectives on the future of Land Surface Models and the challenges of representing complex terrestrial systems. Journal of Advances in Modeling Earth Systems, 0–3. https://doi.org/10.1029/2018ms001453

- 2.18: Maybe clarify whether "fractional vegetation cover" includes all vegetation cover combined (whether grasses or not) or estimates are produced separate for grass, shrub, and "agricultural" vegetation types (as hinted at in 3.1)

**Fractional vegetation cover includes vegetation for specific functional groups, in this case grass, shrubs, or agricultural. The majority of cameras have a single**

vegetation type, but several cameras include both shrubs and grasses and these are separated using different regions of interest and treated as separate time series. We will add the following clarification to the section at 2.18 for clarification:

"Despite its name the PhenoGrass model can theoretically apply to any vegetation type with a distinct growth signal in response to precipitation, as hypothesized in the original threshold-delay model (Ogle  Reynolds 2004). Here we use two other vegetation types, shrubs and agricultural plots, to test how applicable it is beyond grasslands."

Ogle, K.,  Reynolds, J. F. (2004).  Plant responses to precipitation in desert ecosystems: integrating functional types, pulses, thresholds, and delays. Oecologia, 141(2), 282–294. https://doi.org/10.1007/s00442-004-1507-5

- 3.7: It seems that the original Daymet product has a 1-km resolution. Is this a typo or was it aggregated here to 4-km. If the latter, explain why and how.

**This was a typo and the resolution is indeed 1km.**

- 3.7: "Climate time series": daily meteorological time series data rather sound like weather data to me

**We will change this to the following text:**

"For historic precipitation and temperature we used the 1-km resolution Daymet dataset (Thornton et al. 2018), extracting daily time series for the pixel at each PhenoCam tower location."

- 3.11: Why use a 20+ year old global soil dataset instead of using one of the many updated and improved ones, e.g., WISE-based HWSD (Batjes 2016), SoilGrids (Hengl

et al. 2017) etc., and why not use a regional dataset, e.g., gNATSGO (NRCS 2020), POLARIS (Chaney et al. 2019), etc.? How sensitive is the PhenoGrass model to differences in soil variables that occur typically between Global Soil Data Task Group (2000) and others?

**The global soil dataset we used was the one used in Hufkens et al. 2016, and we sought to replicate that as much as possible. Among the datasets suggested here, none have both variables required (field capacity and wilting point), thus we can't make any comparisons with them.**

- 5.15: Why are the different model fits/parametrizations now suddenly called "iterations"? This is confusing because "iteration" can have a specific and different meaning in parameter estimation/model fitting than what appears to be implied here.

**We will remove "iteration" from the text and replace it with just "model", and emphasize how it refers to a parameterization of a specific spatial scale as described above.**

- Fig. 3: Explain what the individual data points are, days pooled from all sites and years?

**Each point is an observed versus predicted daily fCover values from all sites and years within a single spatial scale (see above). We'll change the Fig. 3 and 4 text to clarify that**

**" Figure 3. Observed and predicted daily fCover values of the All Site model and the three vegetation type models, each using all available sites and years with the respective spatial scale. "**

**" Figure 4. Observed and predicted daily fCover values for models from seven spatial scales, where only specific vegetation types within a single ecoregion**

**were used in model fitting. Each uses all available sites and years with the respective spatial scale. "**

References -

Batjes, N. H. 2016. Harmonized soil property values for broad- scale modelling (WISE30sec) with estimates of global soil carbon stocks. Geoderma 269:61–68. https://doi.org/10.1016/j.geoderma.2016.01.034. - Bouchet, P. J., A. T. Peterson, D. Zurell, C. F. Dormann, D. Schoeman, R. E. Ross, P. Snelgrove, A. M. M. Sequeira, M. J. Whittingham, L. Wang, G. Rapacciuolo, S. Oppel, C. Mellin, V. Lauria, P. K. Krishnakumar, A. R. Jones, S. Heinänen, R. K. Heikkinen, E. J. Gregr, A. H. Fielding, M. J. Caley, A. M. Barbosa, A. J. Bamford, H. Lozano-Montes, S. Parnell, S. Wenger, and K. L. Yates. 2019. Better Model Trans- fers Require Knowledge of Mechanisms. Trends in Ecology Evolution 34:489–490. https://doi.org/10.1016/j.tree.2019.04.006.

- Chaney, N. W., B. Minasny, J. D. Herman, T. W. Nauman, C. Brungard, C. L. S. Morgan, A. B. McBratney, E. F. Wood, and Y. T. Yimam. 2019. POLARIS soil properties: 30-meter probabilistic maps of soil properties over the contiguous United States. Water Resources Research 55:2916–2938. https://doi.org/10.1029/2018WR022797.

- Clark, J. S., S. R. Carpenter, M. Barber, S. Collins, A. Dobson, J. A. Foley, D. M. Lodge, M. Pascual, R. Pielke, W. Pizer, C. Pringle, W. V. Reid, K. A. Rose, O. Sala, W. H. Schlesinger, D. H. Wall, and D. Wear. 2001. Ecological Forecasts: An Emerging Imperative. Science 293:657–660. https://doi.org/10.1126/science.293.5530.657.

- Grimm, V., and U. Berger. 2016. Structural realism, emergence, and predictions in next-generation ecological modelling: Synthesis from a special issue. Ecological Modelling 326:177–187. https://doi.org/10.1016/j.ecolmodel.2016.01.001.

- Hengl, T., J. M. de Jesus, G. B. M. Heuvelink, M. R. Gonzalez, M. Kilibarda,

A. BlagoticÌĄ, W. Shangguan, M. N. Wright, X. Geng, B. Bauer-Marschallinger, M. A. Guevara, R. Vargas, R. A. MacMillan, N. H. Batjes, J. G. B. Leenaars, E. Ribeiro, I. Wheeler, S. Mantel, and B. Kempen. 2017. SoilGrids250m: Global gridded soil information based on machine learning. PLOS ONE 12:e0169748. https://doi.org/10.1371/journal.pone.0169748.

- IPCC. 2014. Climate change 2013: the physical science basis. Contribution of working group I to the fifth assessment report of the Intergovernmental Panel on Climate Change. 1535 pages. Cambridge University Press, Cambridge, United Kingdom and New York, NY, USA.

- Pennekamp, F., M. W. Adamson, O. L. Petchey, J.-C. Poggiale, M. Aguiar, B. W. Kooi, D. B. Botkin, and D. L. DeAngelis. 2017. The practice of prediction: What can ecologists learn from applied, ecology-related fields? Ecological Complexity 32:156–167. https://doi.org/10.1016/j.ecocom.2016.12.005.

- Radchuk, V., S. Kramer-Schadt, and V. Grimm. 2019. Transferability of Mechanistic Ecological Models Is About Emergence. Trends in Ecology Evolution 34:487–488. https://doi.org/10.1016/j.tree.2019.01.010. - Soil Survey Staff. 2020. Gridded National Soil Survey Geographic (gNATSGO) Database for the Conterminous United States. Available online at https://nrcs.app.box.com/v/soils. (FY2020 official release). United States Department of Agriculture, Natural Resources Conservation Service.

- White, E. P., G. M. Yenni, S. D. Taylor, E. M. Christensen, E. K. Bledsoe, J. L. Simonis, and S. K. M. Ernest. 2019. Developing an automated iterative near-term forecasting system for an ecological study. Methods in Ecology and Evolution 10:332–344. https://doi.org/10.1111/2041-210X.13104.

- Wiens, J. A. 1989. Spatial Scaling in Ecology. Functional Ecology 3:385–397.

- Yates, K. L., P. J. Bouchet, M. J. Caley, K. Mengersen, C. F. Randin, S. Parnell, A. H. Fielding, A. J. Bamford, S. Ban, A. M. Barbosa, C. F. Dormann, J. Elith, C. B. Embling,

G. N. Ervin, R. Fisher, S. Gould, R. F. Graf, E. J. Gregr, P. N. Halpin, R. K. Heikkinen, S. Heinänen, A. R. Jones, P. K. Krishnakumar, V. Lauria, H. Lozano-Montes, L. Mannocci, C. Mellin, M. B. Mesgaran, E. Moreno-Amat, S. Mormede, E. Novaczek, S. Oppel, G. Ortuño Crespo, A. T. Peterson, G. Rapacciuolo, J. J. Roberts, R. E. Ross, K. L. Scales, D. Schoeman, P. Snelgrove, G. Sundblad, W. Thuiller, L. G. Torres, H. Verbruggen, L. Wang, S. Wenger, M. J. Whittingham, Y. Zharikov, D. Zurell, and A. M. M. Sequeira. 2018. Outstanding Challenges in the Transferability of Ecological Models. Trends in Ecology  Evolution 33:790–802. https://doi.org/10.1016/j.tree.2018.08.001.

---

## Author Response (AR2)

Thank you to the reviewer and editor for this further consideration of our manuscript. Below are the reviewer comments with our replies in bold.

I find the revised manuscript "Multi-scale assessment of a grassland productivity model" to be much improved. I find the new introduction paragraph about "low-dimensional" models interesting and the expanded method section on estimating optimal parameter values (as well as the appendix) helpful.

**Thank you.**

• The authors emphasize in the responses to individual comments of both reviewers that the methods (e.g., model version, the input datasets, model evaluation) were carried out exactly as by Hufkens et al. 2016 – however, this point (and the rationale for why this was done) remains unclear in the manuscript. The method section in the manuscript should spell out that this work exactly re-created the methods used by Hufkens et al. 2016 (in terms of model formulation, input data, evaluation, etc.) but with more data, more sites, and more vegetation types. For instance, the section "2.3 Environmental Data" should mention that these datasets were chosen because these are the ones used by Hufkens et al. 2016 (and not because they are the most adequate), etc.

**We've added text in the method and introduction sections to emphasize this point.**

• I can understand that the authors chose to the Global Soil Data Task Group dataset because it re-creates the Hufkens et al. 2016 analysis – however, this reason needs to be explained in the manuscript (see above). In the response, the authors incorrectly state that none of the datasets suggested by reviewer 1 (see comment to original page 3 line 11) provide required variables (field capacity and wilting point). For instance, both NRCS soil data (gNATSGO, gSSURGO) and POLARIS provide water content at field capacity and wilting point (references in original comment of reviewer 1).

**We apologize for not explaining further how the referenced datasets would not meet our needs. Specifically:**

**SoilGrids**
**According to the FAQ here (https://www.isric.org/explore/soilgrids/faq-soilgrids#Which_soil_properties_are_predicted_by_SoilGrids) there are no variables related to water holding capacity. The primary site (https://soilgrids.org/) also does not have any water related data.**

**WISE Soil Property Databases**
**According to the metadata here (https://www.isric.org/explore/wise-databases) and Batjes 2016 the only water variable is "Available water storage" which we assume is maximum saturated water content.**

**gNATSGO,gSSURGO**
**According to the dataset documentation here (https://www.nrcs.usda.gov/wps/portal/nrcs/detail/soils/survey/geo/?cid=nrcs142p2_053631) , specifically the "SSURGO Metadata – Table Column Descriptions Report" file, the only water variable is "Available Water Storage" at various depths. Which we assume to be maximum saturated water content.**

**POLARIS**
**Using the dataset description here:**
**http://hydrology.cee.duke.edu/POLARIS/PROPERTIES/v1.0/Readme**

**There are two water variables listed:**

**theta_s - saturated soil water content, m3/m3**
**theta_r - residual soil water content, m3/m3**

**Therefore we had to assume that field capacity and wilting point (commonly theta_fc and theta_wp) are not available. Unfortunately the POLARIS journal article (Chaney et al. 2016) does not go into more detail on available variables.**

**If field capacity and wilting point are indeed in these datasets then they should improve their documentation. Regardless we still would have used the original soil dataset for the reason above.**

> Batjes, N.H., 2016. Harmonized soil property values for broad-scale modelling (WISE30sec) with estimates of global soil carbon stocks. *Geoderma*, *269*, pp.61-68.
> Chaney, N.W., Wood, E.F., McBratney, A.B., Hempel, J.W., Nauman, T.W., Brungard, C.W. and Odgers, N.P., 2016. POLARIS: A 30-meter probabilistic soil series map of the contiguous United States. *Geoderma*, *274*, pp.54-67.

• I do not follow the authors' response to reviewer 2 comment asking why "GPP data observed at fluxnet sites" were not used to evaluate PhenoGrass. The authors argue in their response that PhenoGrass predicts fCover and not GPP and that therefore GPP data cannot be used. However, the manuscript asserts that fCover is derived from Gcc by a scaling factor (supplement and page 2 line 23) and that "Gcc … is highly correlated … flux tower derived primary productivity (Yan et al., 2019; Toomey et al., 2015)". So, I do not understand why fCover cannot be compared against observed GPP from fluxnet sites.

**We read the original comment as suggesting using flux tower derived GPP to fit the models, which is not possible with the current model form. Using flux data to further evaluate the model performance though, where Phenocam and flux towers are matched, is a reasonable suggestion and in fact done in Hufkins 2016. Unfortunately it would require considerable more time to incorporate this analysis, but we have included text suggesting this in the discussion.**

• The revised version clarified that R2 was the coefficient of determination and named it unambiguously NSE. The last paragraph of the discussion section draws comparisons with Hufkens et al. 2016 (page 10 lines 6ff) "they had an average R2 of 0.71" – please clarify in the manuscript whether that is NSE or some other variant/interpretation of R2.

**After reviewing the Hufkens 2016 supplement this was also the coefficient of determination and we changed in the text here to NSE to avoid any confusion.**